# Topological quadratic-node semimetal in a photonic microring lattice

Zihe Gao [1] ✉, Haoqi Zhao [2], Tianwei Wu [1], Xilin Feng[2], Zhifeng Zhang[1,2], Xingdu Qiao[2], Ching-Kai Chiu [3] ✉ & Liang Feng [1,2] ✉

Graphene, with its two linearly dispersing Dirac points with opposite windings, is the minimal topological nodal configuration in the hexagonal Brillouin zone. Topological semimetals with higher-order nodes beyond the Dirac points have recently attracted considerable interest due to their rich chiral physics and their potential for the design of next-generation integrated devices. Here we report the experimental realization of the topological semimetal with quadratic nodes in a photonic microring lattice. Our structure hosts a robust second-order node at the center of the Brillouin zone and two Dirac points at the Brillouin zone boundary—the second minimal configuration, next to graphene, that satisfies the Nielsen–Ninomiya theorem. The symmetry-protected quadratic nodal point, together with the Dirac points, leads to the coexistence of massive and massless components in a hybrid chiral particle. This gives rise to unique transport properties, which we demonstrate by directly imaging simultaneous Klein and anti-Klein tunnelling in the microring lattice.

Topological nodal semimetals[1–4], with graphene as the most iconic example[5], possess band touching points protected by symmetries and are characterized by a non-zero quantized topological number[6]. Remarkably, the discovery of graphene[5] ushered in the era of topological nodal semimetals with significant advancements in both fundamental physics and technological innovations. The two Dirac cones in graphene manifest a playground with abundant phenomena ranging from topological physics[7], chiral quantum transport[8], valleytronics[9,10], to moiré bands[11] and unconventional superconductivity[12] in twisted bilayers, just to name a few. Two-dimensional (2D) topological nodal semimetals beyond graphene, particularly the ones with quadratic nodal points beyond linear Dirac points, have been of great interest for their rich quantum-electrodynamics-like chiral physics[8,13–16] or anomalous orbital transport[17]. These quantum transport properties are fundamentally different from those in graphene, as a direct result of the difference in topological node configuration, and are potentially foundations for next-generation electronic and photonic devices. For example, massive chiral particles hosted by the quadratic nodal point have been a promising platform for the realization of 2D field effect transistors (FETs), impossible in graphene due to the linear Dirac

dispersion and the Klein tunneling[8]. Theorists have proposed potential realizations of quadratic point contacts in 2D and three-dimensional (3D) semimetals[17–19]. However, although quadratic nodes can emerge in a bilayer graphene platform with fine-tuning[16,20], quadratic nodes with topological protection have remained largely elusive in condensed-matter materials despite myriad efforts due to the lack of spatial inversion symmetry in gated bilayer graphene[21,22] and the lack of isolated orbital bands in orbitronics[17]. Photonic and acoustic metamaterials, on the other hand, have provided versatile platforms where quadratic band touching points have been demonstrated in 2D photonic crystals and topologically protected high-order band touching points have been demonstrated in 3D Weyl system[23–26].

Here we report the experimental realization of the topological quadratic-node semimetal, with coexisting symmetry-protected quadratic and Dirac nodes, distinct from any previously reported nodal systems. The coexistence of quadratic and Dirac nodes, rooted in the cross-spin coupling and the commutation relations between symmetry operators, manifests profoundly the unique chiral transport. The quadratic-node semimetal is constructed on a triangular lattice with winding cross-spin coupling between nearest-

[1]Department of Materials Science and Engineering, University of Pennsylvania, Philadelphia, PA 19104, USA. [2]Department of Electrical and Systems Engineering, University of Pennsylvania, Philadelphia, PA 19104, USA. [3]RIKEN Interdisciplinary Theoretical and Mathematical Sciences (iTHEMS), Wako, Saitama 351-0198, Japan. ✉e-mail: zihegao@seas.upenn.edu; ching-kai.chiu@riken.jp; fenglia@seas.upenn.edu

neighbouring sites, experimentally realized in a photonic lattice with degenerate clockwise (CW) and counter-clockwise (CCW) modes constituting the pseudospin space (Fig. 1). The cross-spin coupling is a result of momentum conservation during the coupling process between adjacent rings (depicted in Fig. 1b and quantitatively analyzed in Supplementary Information Section 10). Similar to graphene, the quadratic-node semimetal is also described by a two-band tight-binding model, and it obeys the Nielsen−Ninomiya theorem[27−29]. For graphene, with winding number ±1 at the two inequivalent cones respectively, it possesses the minimal topological node configuration in the hexagonal Brillouin zone and naturally obeys the Nielsen−Ninomiya theorem, which forces the total topological number in the Brillouin zone to be neutral[27−29]. The quadratic-node semimetal we report here is the second minimal configuration according to the generalized Nielsen−Ninomiya theorem[30], with a winding number of +2 at Γ and two Dirac nodes at K and K′ with a winding number of −1, illustrated in Fig. 1c. With an extra valley at the center of the Brillouin zone, the quadratic-node semimetal exhibits an extra dimension in its valley degree of freedom beyond graphene, and the transport in this extra valley can be modulated independently to the Dirac valleys. With edge excitation, a hybrid chiral particle can be excited, which simultaneously exhibits massive and massless behaviors, opening possibilities for exotic dynamics. This unique nodal configuration, with coexisting quadratic and Dirac nodes around the same energy level, is distinct from other topological photonic or acoustic metamaterials and presents a versatile three-valley system. The quadratic-node semimetal inherits graphene physics, with massless Dirac particle and

Klein tunneling supported by its linear nodal points, but at the same time features a new dimensionality represented by its quadratic node. It can enable a distinguished, promising platform to study topological phases and bilayer coupling beyond graphene by exploring various ways in gapping and coupling between cones and benefiting from the vast possibilities offered by the extra quadratic cone. From the technological perspective, the robust quadratic node and its associated massive chiral particle also open new possibilities for building photonic FETs using anti-Klein tunneling[8,31].

## Results

### 2D quadratic-node semimetal and its topological node configuration

The quadratic-node semimetal, using the weakly-coupled photonic microring lattice with whispering gallery mode (WGM) order 34 as an example, can be described by a 2 × 2 tight-binding model[30] (see Supplementary Information Section 1)

$$\hat{H} = -\sum_{\tilde{x},\tilde{y}} \left\{ \hat{C}^{\dagger}_{\tilde{x}+1,\tilde{y}} \sigma_x \hat{C}_{\tilde{x},\tilde{y}} + \hat{C}^{\dagger}_{\tilde{x}-\alpha,\tilde{y}+\gamma}(\alpha\sigma_x + \gamma\sigma_y)\hat{C}_{\tilde{x},\tilde{y}} \right.$$
$$\left. + \hat{C}^{\dagger}_{\tilde{x}+\alpha,\tilde{y}+\gamma}(\alpha\sigma_x - \gamma\sigma_y)\hat{C}_{\tilde{x},\tilde{y}} \right\} + h.c., \quad (1)$$

where lattice constant ≡ 1, $\alpha = \cos 2\pi/3$, and $\gamma = \sin 2\pi/3$, and $\hat{C}_{\tilde{x},\tilde{y}} = \begin{pmatrix} \hat{c}_{\circlearrowleft_{\tilde{x},\tilde{y}}} \\ \hat{c}_{\circlearrowright_{\tilde{x},\tilde{y}}} \end{pmatrix}$ is the annihilation operator with CCW and CW modes (i.e., the two pseudospins) at lattice site $(\tilde{x},\tilde{y})$. The coupling strength (i.e., hopping

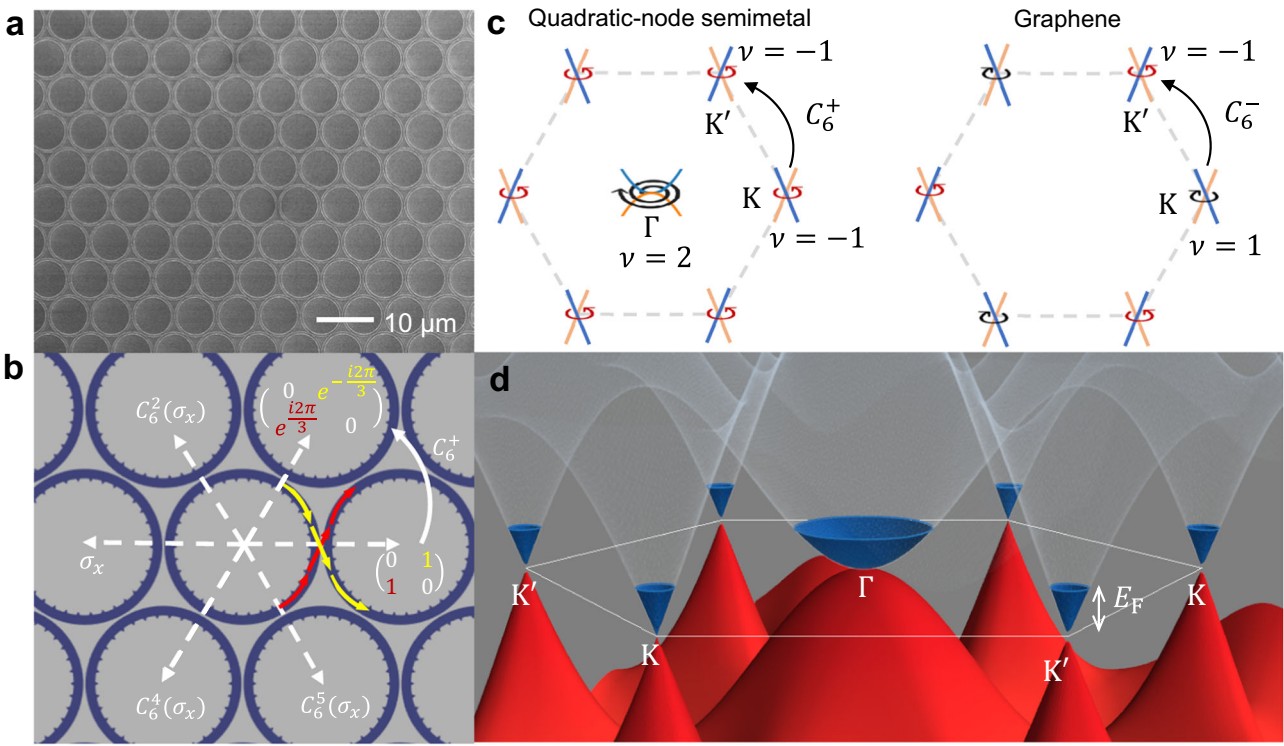

**Fig. 1 | The quadratic-node semimetal realized in a photonic microring lattice. a** Scanning electron microscopy (SEM) image of the microring array in a triangular lattice fabricated on a silicon-on-insulator (SOI) substrate. **b** Cross-spin coupling geometry between the nearest neighbors. The off-diagonal elements in the coupling matrix represent cross-spin coupling, illustrated by the red and yellow arrows. The coupling matrices in the 6 different coupling directions are related by the rotation operator $C_6$ (see Supplementary Information Section 1). **c** Nodal points in the Brillouin zone of the quadratic-node semimetal and the graphene. Γ, K, and K′ mark the high-symmetry points in the Brillouin zone. The + (−) sign on $C_6$ operator denotes whether it commutes (or anticommutes) with the chiral symmetry

operator and hence resulting in nodal points with the same (opposite) windings. The positive (negative) winding numbers, $v$, carried by nodal points, are illustrated by the black (red) winding arrows. At Γ point in the quadratic-node semimetal, the arrow winds twice to illustrate the charge-2 winding carried by the quadratic node. The blue and orange lines illustrate the energy dispersion at the nodal points, which is parabolic when |$v$| = 2 and linear when |$v$| = 1. The blue branch is with pseudospin $(1,-1)^T$ while the orange branch is with pseudospin $(1,1)^T$. **d** The energy dispersion of the quadratic-node semimetal, with an example Fermi energy, $E_F$, of 0.5 in the unit of normalized coupling strength. The red (blue) color denotes the negative (positive) energy branch.

integral) is set to unity. The key ingredient in this Hamiltonian is the cross-spin coupling between nearest neighbors, represented by the coupling matrix consisting of superpositions of Pauli matrices $\sigma_x = \begin{pmatrix} 0 & 1 \\ 1 & 0 \end{pmatrix}$ and $\sigma_y = \begin{pmatrix} 0 & -i \\ i & 0 \end{pmatrix}$, naturally arising from the conservation of momentum in the coupling region between two adjacent microrings (Fig. 1b). This coupling matrix has direction dependence and respects $C_6$ symmetry: $\kappa_\beta$, the coupling matrix along direction $\beta$, satisfies $\kappa_{C_6(\beta)} \equiv \kappa_{\beta+\pi/3} = C_6(\kappa_\beta)$ (see Supplementary Information Section 1). Note that the cross-spin coupling geometry differentiates our photonic lattice from previously studied topological photonic lattices where the two pseudospins do not cross couple (and hence can be treated simply as two time-reversed copies of the same system)[32,33]. The phase in the off-diagonal terms of the coupling matrix is dependent on the phase profile of the resonant mode along the ring perimeter and hence the WGM mode order (see Supplementary Information Section 1). For mode order 34, the coupling respects $C_6$ symmetry with the rotation operator defined as $C_6 = \begin{pmatrix} e^{i2\pi/3} & 0 \\ 0 & e^{-i2\pi/3} \end{pmatrix}$. In the momentum space, the Hamiltonian is rewritten as

$$H(\mathbf{k}) = \begin{pmatrix} 0 & h(\mathbf{k}) \\ h^*(\mathbf{k}) & 0 \end{pmatrix}, \qquad (2)$$

where $h(\mathbf{k}) = 2[\cos k_x + \cos(\alpha k_x + \gamma k_y) + \cos(-\alpha k_x + \gamma k_y)]$. Due to the geometry of the lattice, the Hamiltonian naturally preserves inversion symmetry ($H(-\mathbf{k}) = H(\mathbf{k})$). In addition, because the two pseudospins (CCW and CW modes) are time-reversal partners, time-reversal symmetry is preserved ($\sigma_x H^*(-\mathbf{k})\sigma_x = H(\mathbf{k})$), where $\sigma_x$ exchanges the pseudospins. Hence, space-time inversion symmetry is preserved ($\sigma_x H^*(\mathbf{k})\sigma_x = H(\mathbf{k})$) so that $\sigma_z$ is forbidden and the nodal points are therefore symmetry-protected. In this two-band model, this symmetry is equivalent to chiral symmetry ($\sigma_z H(\mathbf{k})\sigma_z = -H(\mathbf{k})$), regardless of $\sigma_0$ term. Due to chiral symmetry, the off-diagonal term $h(\mathbf{k})$ can be used to compute the winding number $\nu = \frac{i}{2\pi} \oint \nabla_\mathbf{k} (\ln \det[h(\mathbf{k})]) \cdot d\mathbf{k}$, where the integral path encloses the node[34]. The non-zero winding number characterizes non-trivial nodal points: $\nu = \pm 1$ corresponds to a Dirac node, and $\nu = \pm 2$ is associated with a quadratic node.

As chiral symmetry (with the symmetry operator $S = \sigma_z$) quantizes the winding numbers of the nodes, the commutation relations between $S$ and other symmetry operators differentiate the topological nodes between the quadratic-node semimetal and the graphene, illustrated in Fig. 1c. In both systems, the Dirac cones at K, K′ are connected by $C_6$ rotation symmetry (or mirror symmetry, time-reversal symmetry). We use this rotation symmetry operator as an example to discuss the winding numbers of the nodes. Since for graphene, the $C_6$ rotation operator, which exchanges the sublattice bases, anticommutes with the chiral (sublattice) operator, the winding numbers at these two points are opposite $\nu(K) = -\nu(K')$. In contrast, because the pseudospin in the quadratic-node semimetal is not spatial dependent, the rotation operator naturally commutes with the chiral operator; hence, the nodes at K, K′ have the same winding numbers. According to the Nielsen–Ninomiya theorem[27–30], the charge neutralization must lead to the presence of at least a nodal point with winding number 2 in the same Brillouin zone. Since the $C_6$ rotation symmetry can duplicate the nodal point away from the rotation centers, the presence of a quadratic node at Γ, the only $C_6$ rotation center, forms the minimal configuration.

The band structure calculated from the tight-binding Hamiltonian is plotted in Fig. 1d with an example Fermi level set to 0.5 (in unit of the normalized coupling strength). It is expected from the band filling that the quantum transport properties, determined by the low energy physics, can carry richer dynamics than graphene due to the coexistence of quadratic and Dirac nodal points. Before the study of transport, we first provide a direct characterization of the topological

charges of the two different types of nodal points. Near the Dirac nodes, the quasiparticles obey effective Hamiltonian

$$\begin{aligned} H_{K,K'} &= \pm \frac{3\sqrt{3}}{2} \begin{pmatrix} 0 & \Delta k_x + i\Delta k_y \\ \Delta k_x - i\Delta k_y & 0 \end{pmatrix} \\ &= \pm \frac{3\sqrt{3}}{2} \begin{pmatrix} 0 & ke^{i\theta} \\ ke^{-i\theta} & 0 \end{pmatrix} = \pm \hbar v_F \boldsymbol{\sigma} \cdot \Delta\mathbf{k} \end{aligned} \qquad (3)$$

where $\Delta\mathbf{k} = (\Delta k_x, \Delta k_y)$ is the momentum deviation from the high symmetry point (K and K′), $\theta = \tan^{-1}(\frac{\Delta k_y}{\Delta k_x})$, $\boldsymbol{\sigma} = (\sigma_x, \sigma_y)$, and the $\pm$ sign corresponds to K and K′ points, respectively. This 2D Dirac-like equation leads to the well-known massless chiral particle with its spin and momentum locked to parallel or antiparallel direction. Notably, although the two cones in K and K′ are flipped from the $\pm$ sign, the windings in K and K′ are the same, distinguished from the case in graphene. Near the quadratic node, the effective Hamiltonian is

$$H_\Gamma = \frac{3}{4} \begin{pmatrix} 0 & (k_y + ik_x)^2 \\ (k_y - ik_x)^2 & 0 \end{pmatrix} = -\frac{3}{4} \begin{pmatrix} 0 & k^2 e^{-2i\theta} \\ k^2 e^{2i\theta} & 0 \end{pmatrix} \qquad (4)$$

suggesting a chiral particle with finite mass as a consequence of the quadratic dispersion. However, it is important to note that this chiral particle is intrinsically different from massive spinless particles described by the Schrödinger equation. For this massive chiral particle, its spin is again locked to its momentum direction (for each spin, there are two allowed momentum directions instead of one in the massless case), and its transport properties in a homogeneous material or a heterogeneous junction are dominated by its chirality, which is another manifestation of the Klein "paradox" in condensed-matter systems[8], often coined anti-Klein tunneling[15,16,35]. Here we first focus on a homogeneous semimetal and characterize its winding numbers around the Dirac and quadratic nodal points, shown in Fig. 2. The topological charges carried by the nodal points can be observed from the winding of $\phi$, the relative phase between two pseudospin components in the eigenmodes, equivalent to the winding of (pseudo) spin direction along x-y plane on the Bloch sphere. We plot the phase $\phi$ for the top ($E > 0$) branches in Fig. 2a and b for mode order 34 and 35, respectively. For mode order 34, the Hamiltonian, as discussed above, is constructed from the coupling matrix $\kappa_0 = \sigma_x, C_6 = \begin{pmatrix} e^{i2\pi/3} & 0 \\ 0 & e^{-i2\pi/3} \end{pmatrix}$. For mode order 35, the coupling phase is different from mode order 34 due to the difference in the phase profile along the perimeter of a microring resonator, it has $\kappa_0 = -\sigma_x$ and $C_6 = \begin{pmatrix} e^{i\pi/3} & 0 \\ 0 & e^{-i\pi/3} \end{pmatrix}$. These two mode orders thus exhibit opposite phase windings, evidently shown in Fig. 2a and b. To interact with resonant modes in the photonic lattice and perform excitation or imaging from free space, we engrave angular gratings on the inner sidewalls of the rings to couple the optical modes confined in the photonic semimetal to propagating modes in the free space through a collective scattering process (Fig. 2c). The angular gratings scatter CW (CCW) WGMs into right- (left-) handed circularly polarized light, respectively[36]. This maps the pseudospin Hilbert space in the lattice to polarization Hilbert space in the free space: pseudospin $|+\rangle = (1,1)^T$ is mapped to vertical polarization, while pseudospin $|-\rangle = (1, -1)^T$ is mapped to horizontal polarization (see Supplementary Information Section 6). This mapping can be freely controlled by the angular locations of the scatters, which is an advantage of our system. When the number of scatters per resonator (i.e., grating order) does not equal the mode order, the scatters will not affect the effective Hamiltonian of the lattice. The polarization rotation around a quadratic and a Dirac nodal point is calculated and plotted in Fig. 2d and e, showing winding of two cycles and one cycle,

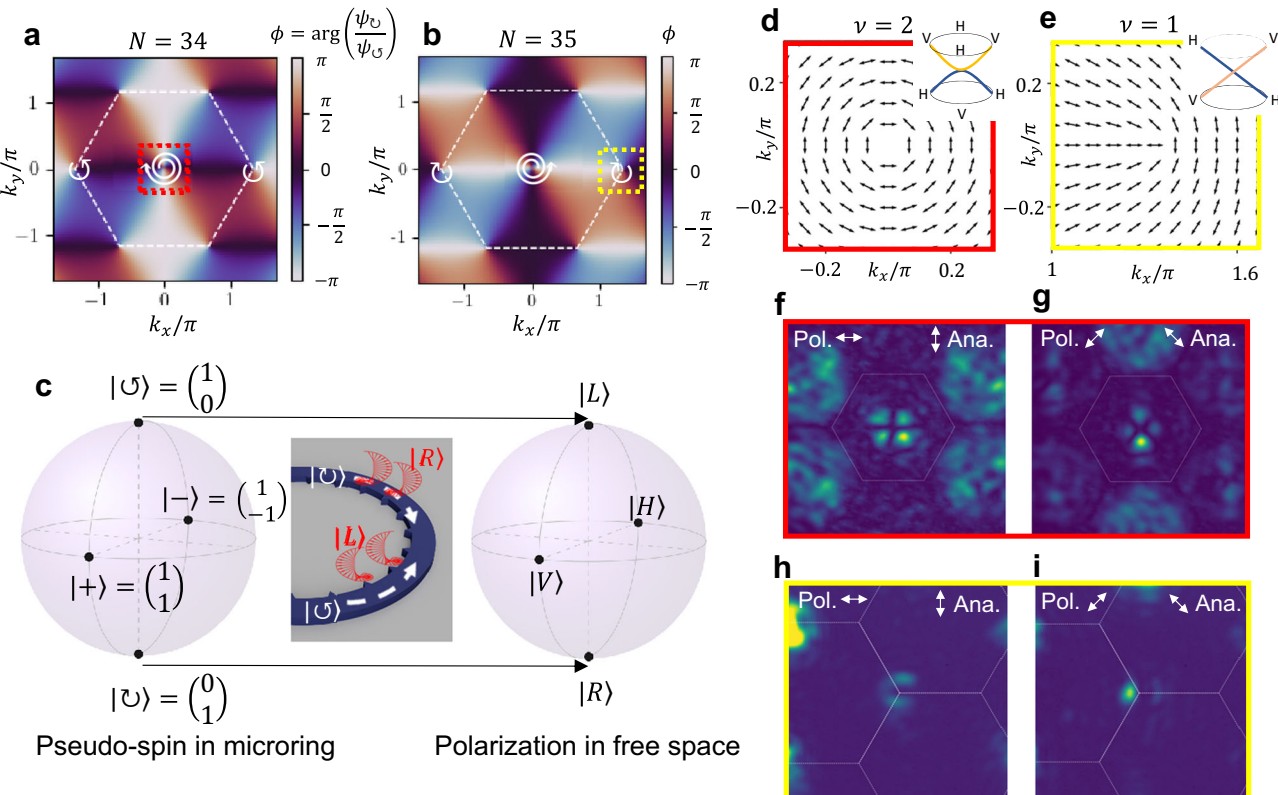

**Fig. 2 | Characterization of the topological charges around the band touching points in the quadratic-node semimetal. a, b** Winding of the wavefunction (pseudo) spin direction, $\phi \equiv \arg(\psi_{\circlearrowright}/\psi_{\circlearrowleft})$, for the positive energy ($E > 0$) branches of WGM order 34 and 35, respectively, in the Fourier space marked by wavevectors $k_{x,y}$. The pseudospin is noted as $(\psi_{\circlearrowright}, \psi_{\circlearrowleft})^T$, where $\psi_{\circlearrowright}$ ($\psi_{\circlearrowleft}$) is the wavefunction component in the CCW (CW) WGM order. The first Brillouin zone is marked by dashed white lines. The charge-1 and charge-2 windings of $\phi$ at the high-symmetry points are marked by white winding arrows. **c** Correspondence between the Bloch sphere representing the pseudospin states in the microrings and the Bloch sphere representing polarization states of the light in free space. $|\circlearrowright\rangle$ and $|\circlearrowleft\rangle$ denote the CW and CCW WGM mode, which we choose as pole states of the pseudospin Bloch sphere. $|+\rangle = (1,1)^T$ and $|-\rangle = (1, -1)^T$ denote the pseudospins in the $\pm x$ directions. $|R\rangle$, $|L\rangle$, $|V\rangle$, $|H\rangle$ are the right-handed, left-handed, vertical, and horizontal polarizations, respectively, in the polarization Hilbert space. This correspondence originates from the spin-direction locking in the scattering process, illustrated in the middle, where the CW (CCW) traveling wave scatters only the right (left) handed circularly polarized light (represented by rotating red arrows). **d, e** Polarization of the scattered light, plotted near the quadratic band touching point (of mode order 34, red dotted box in **a**) and the Dirac band touching point (of mode order 35, yellow dotted box in **b**), respectively ($E > 0$ branches). $\nu$ is the winding number. The polarization rotates two full cycles near the quadratic band touching and one full cycle near the Dirac band touching. Insets illustrate the cones, where orange and blue denote the vertical and horizontal polarizations (marked by V and H, corresponding to $(1,1)^T$ and $(1,-1)^T$ pseudospins in the microrings). **f–i** Experimental Fourier plane images of the cross-polarization reflection off a quadratic-node semimetal with an incident wavelength slightly above the band-touching energy of mode order 34 (**f** and **g**) and mode order 35 (**h** and **i**). Orientations of the polarizer (Pol.) and analyzer (Ana.) used for cross-polarization filtering are labeled on top of the images. The Brillouin zones are outlined with white lines. Images with continuous rotations of the polarizer and analyzer are recorded in Supplementary Movies 1 and 2, where continuous winding of the wavefunction is observed. The measurement setup and a detailed account of the noise (stray light) due to imperfect cross-polarization filtering, for example, near the Dirac cones and in Brillouin zones away from the center one, is discussed in Supplementary Information Section 7.

respectively, in the polarization Hilbert space. Experimentally, this winding is observed with cross-polarization reflection measurement in the Fourier plane. We excite the sample with linearly polarized monochromatic light, and as a result of cross-polarization filtering, dark lines appear in the Fourier domain where the lattice Bloch modes have polarization either identical or orthogonal to the incident polarization (Fig. 2f–i). For the quadratic band touching, two dark lines appear, and they rotate by the same angle as the rotation of polarizer and analyzer (Fig. 2f and g and Supplementary Movie 1). For the Dirac points, we observe a single dark line, which rotates by twice the angle as the rotation of polarizer and analyzer (Fig. 2h and i and Supplementary Movie 2), and the rotation direction is the same for both K and K′. The number of dark lines and the rotation of pseudospin provide a direct measurement of quantized winding numbers of 2 and 1 for the quadratic and Dirac nodal points. Continuous wavelength sweeping (Supplementary Movie 3) also confirms the touching of the top and bottom bands (within the energy resolution limited by the quality factor of individual resonators). With these results, we report

the experimental realization of the topological quadratic-node semimetal, which is a 2D system that possesses a topologically protected quadratic node and obeys the Nielsen–Ninomiya theorem.

## Quantum transport in the quadratic-node semimetal

Near the band touching energy, the quadratic-node semimetal supports both massive and massless chiral particles in three inequivalent valleys in the momentum space, leading to unique quantum transport properties. To study the exotic chiral transport in our quadratic-node semimetal, we first place coherent excitation on the edge of a homogeneous nanoribbon and terminate the nanoribbon with an absorber section (Fig. 3a and Methods). With an in-phase excitation from the waveguides on the edge, we excite the $(1,1)^T$ pseudospin (orange-colored in Fig. 3) with $k_y \cong 0$ (see Supplementary Information Section 4). The band structure along $k_y = 0$ in mode order 34 and 35 are the same, but the pseudospin is flipped (shown in Fig. 3b), because the nearest-neighbor coupling in the horizontal direction, $\kappa_0$, has opposite signs for mode orders 34 ($\kappa_0 = \sigma_x$) and mode order 35 ($\kappa_0 = -\sigma_x$) (see

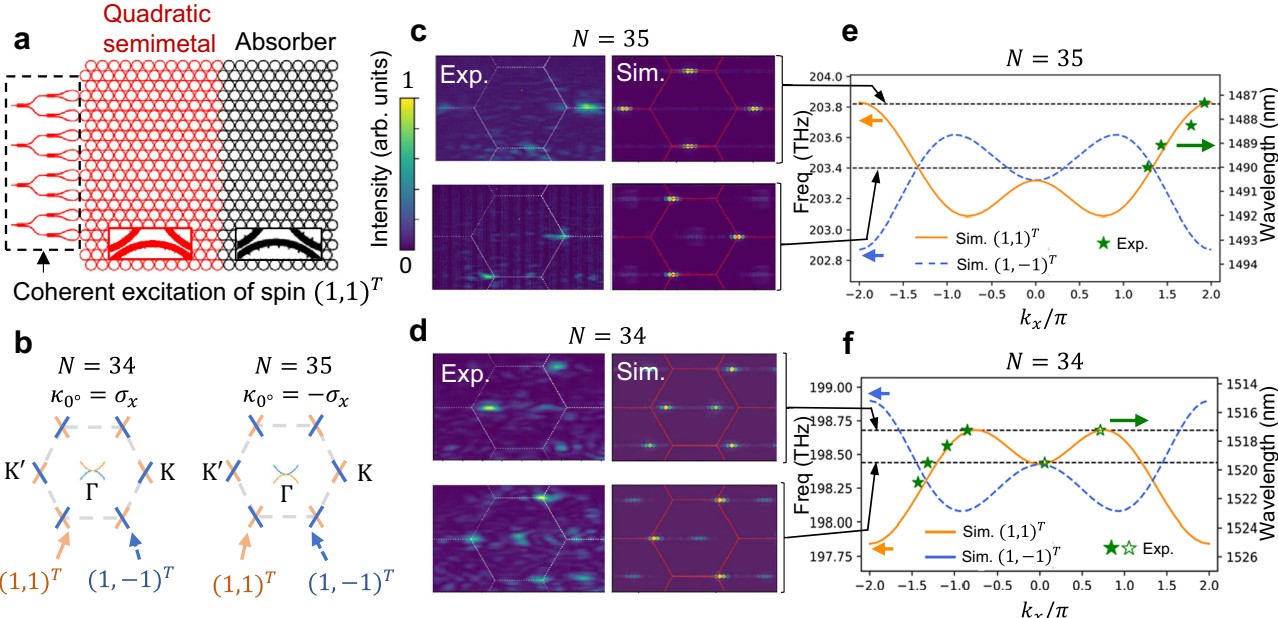

**Fig. 3 | The chiral quantum transport demonstrating valley selectivity and hybrid chiral particles that are simultaneously massive and massless.**
**a** Schematic illustration of the excitation, the homogeneous section (red) and the absorber section (black) in the nanoribbon. The excitation is set to $(1,1)^T$ pseudospin (Methods). The absorber section has excessive scattering loss that gradually increases from left to right to eliminate any reflection off the sample boundary. **b** The pseudospin composition of all the band touching points (along cut planes of constant $k_y$). The coupling matrix along $x$-direction, $\kappa_{0°}$, has opposite signs for mode order 34 and 35, resulting in the difference in pseudospin composition. $\sigma_x = \begin{pmatrix} 0 & 1 \\ 1 & 0 \end{pmatrix}$ is the Pauli matrix. **c**, **d** Experimental measurements (Exp.) and finite-element-method (FEM) simulations (Sim.) of the transporting wave excited by the coherent excitation on the edge for mode order 35 and 34, respectively, in the Fourier (reciprocal) space. **c**, **d** share the same normalized intensity colorbar. In the experimental images for mode order 35, the signals are stronger in the lower half of the

Brillouin zone due to the emission profile of the individual rings. This is discussed in detail in Supplementary Information Section 6. **e**, **f** FEM simulation of the energy dispersion (orange and blue lines, left y-axis) and experimentally measured momentum values (green stars) at various wavelengths (right y-axis). The solid (hollow) green stars show locations of the highest (second highest) peaks in the Fourier-space image along $k_y = 0$. The black lines indicate the energy levels for the images shown in c and d. Experimentally for mode order 34 around 1519.5 nm we observe chiral particles that are simultaneously massive and massless, occupying both Dirac and quadratic valleys. Fourier plane images and the protocol in plotting the experimental data points (green stars) and FEM simulations are shown in Supplementary Information Section 8 and Supplementary Movies 6 and 7. The energy offset between the quadratic and the Dirac band touching points is a result of the strong coupling between the microrings (see Supplementary Information Section 5). This reduces the energy range within which we can inject both massive and massless components, as compared to the tight-binding model calculations in Fig. 1 and Supplementary Figs. 3–10.

Supplementary Information Section 1). For mode order 35, above the quadratic-node band touching energy, the $(1,1)^T$ pseudospin excites only the Dirac valley at K with no excitation of the K′ or Γ valleys, shown in Fig. 3c and e, because in K′ and Γ valleys the right-propagating mode has the orthogonal pseudospin $(1,-1)^T$. This valley selectivity from pseudospin conservation is a direct consequence of chirality and spin-momentum locking in both Dirac and quadratic valleys. The spin-momentum locking in our systems is in direct analogy to the spin-momentum locking well known, for example, in topological insulators with the Rashba term:[37] the particle's spin determines the direction of its momentum. Here, a valley can be excited only when the spin-momentum locking is satisfied in this valley. For mode order 34, with similar valley selectivity, we observed excitation of both the Dirac valley and the quadratic valley above the quadratic band touching energy (Fig. 3d and f). In other words, each photon, injected through the spatially localized edge excitation, occupies both the Dirac and the quadratic valleys and becomes a hybrid superposition of both massive and massless chiral particles. This counterintuitive behavior can be explored to show exotic transport properties. As an example, we demonstrate the simultaneous Klein and anti-Klein tunneling as a signature of this hybrid chiral particle.

When a hybrid chiral particle occupies both the quadratic valley and the Dirac valley, the two valley components behave in distinct manners when we apply potential modulation to form a p-n junction (Fig. 4a). Here, the microrings in the n-region have a nominal width of 400 nm, while the ones in the p-region have a width of 395 nm. The difference in resonant energies due to the width difference creates a

potential step of about 350 GHz (about twice the coupling strength). The hybrid chiral particle in this p-n junction nanoribbon simultaneously reveals two signature behaviors of chiral tunneling, the Klein and anti-Klein tunneling, because of the delicately designed coexistence of both Dirac and quadratic valleys. The transport of a massless chiral particle in a Dirac valley at normal incidence on a potential step follows the Klein tunneling mechanism (Fig. 4b)[8,38], originally used to describe the tunneling of relativistic electrons in the Klein "paradox"[39]. The spin-momentum locking in a massless chiral particle forbids the reflection process that violates spin conservation, hence forcing the transmission coefficient to be unity when intervalley scattering is negligible. On the contrary, for the massive chiral particle in a quadratic valley, because for each spin there are two allowed momentum directions, reflection is allowed at normal incidence. When the Fermi energy is lower than the potential barrier, the p-n junction spectrally aligns the massive "electrons" in the n-region to the "holes" in the p-region, and because the "electrons" and "holes" have orthogonal spins, transmission is forbidden by spin conservation. This complete reflection at normal incidence, or known as anti-Klein tunneling, is a complementary example to Klein tunneling and a promising candidate for implementing FETs in 2D semimetals[8]. Previously the anti-Klein tunning has only been observed indirectly in bilayer graphene with conductance measurements[15,16,31] without direct information on the wavefunction or the reflection process. Here we provide not only direct observation of the anti-Klein tunneling in the Fourier plane, thanks to the topologically protected quadratic cone, but also we show that a

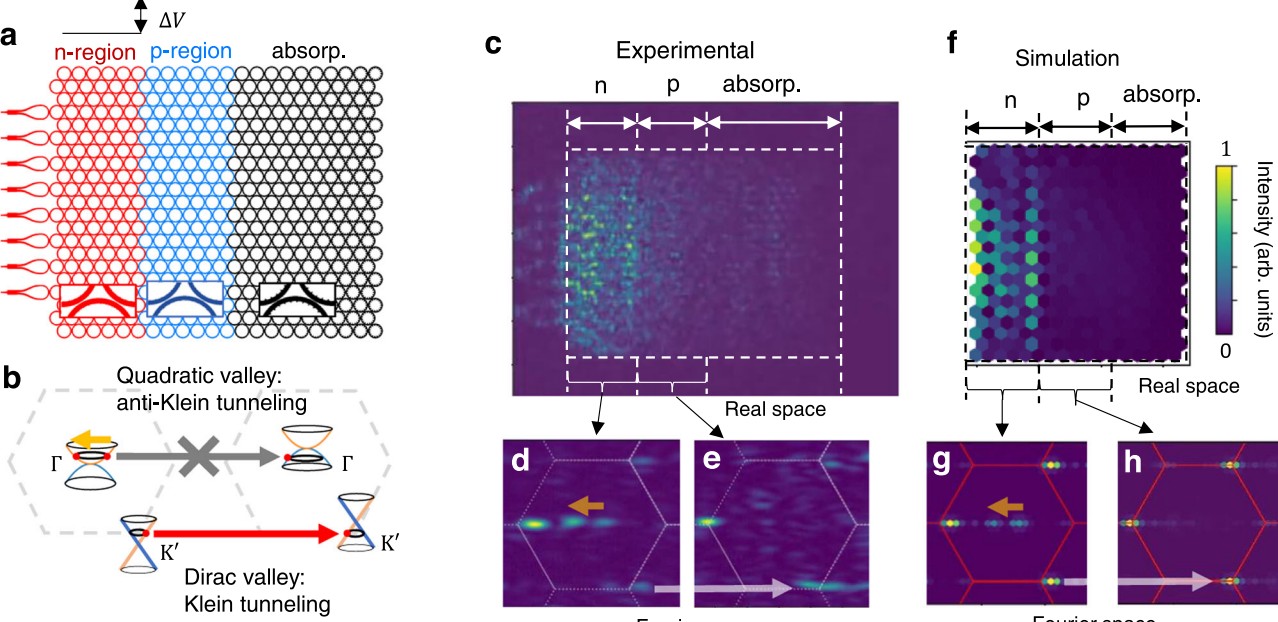

**Fig. 4 | The simultaneous Klein and anti-Klein tunneling for a hybrid chiral particle in a p-n junction. a** Schematic of the p-n junction, where the p-region (blue) consists of narrower rings (hence higher resonant energy) compared to the n-region (red), creating a potential step $\Delta V$. The absorption region (absorp., marked by black) absorbs any outgoing wave from the end of the p-region. **b** Illustration of simultaneous Klein tunneling in the K′ valley and anti-Klein tunneling in the Γ valley in a quadratic-node semimetal p-n junction. The Klein tunneling features complete transmission, as the transmission process (red arrow) conserves the pseudospin, while the reflection process requires pseudospin-flipping. The anti-Klein tunneling features complete reflection, as the reflection process conserves the pseudospin (yellow arrow), while the transmission process requires pseudospin-flipping. **c**–**h** Experimental measurement and FEM simulation

of the chiral transport with coherent edge excitation and hybrid chiral particles. **c** and **f** are the wavefunction intensity distribution in the sample plane. The n, p, and absorber regions are labeled. Both transmission and reflection can be observed at the junction from the coexisting Klein and anti-Klein tunneling. **d** and **e** are the Fourier plane images captured with an aperture allowing only the n- (p-) region to be imaged (Methods). The n-region shows reflection (two intensity peaks near the Γ point, marked by the yellow arrow, with group velocities in the +x and -x directions, respectively) from the anti-Klein tunneling near the quadratic valley and the p-region shows transmission (white arrow) near the K′ Dirac valley, agreeing well with the simulation result shown in **g** and **h**. Additional Fourier plane images and simulations are shown in Supplementary Information Section 9.

hybrid chiral particle can exhibit both Klein and anti-Klein tunneling. The simultaneous Klein and anti-Klein tunneling processes at Γ and K′ valleys are illustrated in Fig. 4b. Tight-binding model calculation demonstrating unitary reflection (transmission) for each valley is shown in Supplementary Information Section 2 and Supplementary Movies 4 and 5 (rich dynamics involving simultaneous Klein and anti-Klein tunneling can be observed in the time domain simulation in Supplementary Movies 4 and 5). Nanoribbons are also theoretically modeled, showing high reflection (transmission) albeit imperfect collimation limited by the finite nanoribbon width (see Supplementary Information Section 3). Experimentally we observe both reflection and transmission at the p-n junction in the sample plane (Fig. 4c), which is a combined effect from both the Klein and anti-Klein tunneling (Fig. 4f and Supplementary Movie 8). In the Fourier plane, we can directly observe the momentum composition of the wavefunction in the n- and p-regions, respectively, with a spatial filter placed at the image plane (Methods). In the n-region (Fig. 4d and g), for the massive chiral component in the quadratic valley, both the incident and reflected wave occur with opposite momenta, as a direct observation of the anti-Klein tunneling. On the other hand, for the massless chiral component in the Dirac valley, incident wave transmits without any reflection, as predicted by Klein tunneling. In the p-region (Fig. 4e and h), transmission appears in the Dirac valley only, with no transmission in the quadratic valley. The direct imaging of simultaneous Klein and anti-Klein tunneling demonstrates the unique transport property in the quadratic-node semimetal as a result of coexisting massive and massless components in the chiral particle. Massive chiral particle in 2D semimetals is a promising approach to create logic gates. The simultaneous Klein and anti-Klein

tunneling in two separate valleys, on the other hand, provide two channels that can be independently modulated.

## Discussion

Here we report the discovery of a sibling of graphene, the topological quadratic-node semimetal. The quadratic-node semimetal is described by a two-band tight-binding model, much like the model of graphene, except for the different nodal configurations (Supplementary Table 1), which is governed by the Nielsen–Ninomiya theorem. The theorem implies that the quadratic node is accompanied by two Dirac nodes. The coexistence of quadratic and Dirac nodes, rooted in the cross-spin coupling and the commutation relations between symmetry operators, manifests profoundly the unique transport that we report here, as both massive and massless chiral components are hosted and they support distinct tunneling behaviors due to their difference in spin-momentum locking. We show direct characterizations of the nodal topological charges and more importantly, the quantum transport showing that a chiral particle can be simultaneously massive and massless, exhibiting both Klein and anti-Klein tunneling. The quadratic-node semimetal is distinct from any previously reported condensed-matter materials, as well as photonic or acoustic metamaterials (compared and summarized in Supplementary Section 11 and Supplementary Table 1), and it presents an intriguing case for quantum-mechanical studies. With an extra valley at the center of the Brillouin zone, the quadratic-node semimetal exhibits an extra dimension in its valley degree of freedom compared to graphene, and the transmission in the extra valley can be modulated independently to the Dirac valley. From the technological perspective, with three valleys and numerous ways to gap them, the quadratic-node semimetal can be explored as a new foundation in

building integrated nonreciprocal topological devices[40,41] and may especially benefit applications harnessing inter-valley scattering dynamics[42] or valley-dependent optical effects[43]. In photonics, the spin-momentum locking can be used for in-plane beam steering, and with the capability of modulating the refractive index and hence the resonance frequencies of the resonators (i.e., effective doping), the metamaterial can support transport channels and spin-valley dynamics tunable in real-time with potential applications in communications, information processing, power transfer, to name a few.

## Methods

### Device design and fabrication

The samples were fabricated on a silicon-on-insulator (SOI) substrate with a 220-nm-thick silicon layer and silicon nitride cladding on top. The homogeneous photonic semimetal characterized by cross-polarization reflection (Fig. 3 and Supplementary Movies 1–3) in design has a ring outer radius of 3.65 μm, width of 405 nm, angular grating scatter size of 180 nm, and scatter order of 33 (so that WGM order 34 has strong emission at the center of Brillouin zone, while order 35 has strong emission at the first Brillouin zone boundary, see Supplementary Section 6).

The homogeneous photonic semimetal nanoribbon in design has a ring outer radius of 3.55 μm, width of 445 nm, angular grating scatter size of 80 nm, and scatter order of 33. The scatter size for the absorption layer gradually increases from 80 nm to 180 nm to provide dissipation by excessive scattering to the free space. The excitation was implemented with a grating coupler and four stages of multi-mode-interferometers (MMIs) shown in Supplementary Fig. 17b, which split the single input waveguide into $2^4$ waveguides that are connected to the microrings on edge evanescently. Because each of the $2^4$ waveguides have identical optical path, this excitation excites the fundamental mode with $(1,1)^T$ pseudospin.

The photonic semimetal nanoribbon p-n junction in design has a ring outer radius of 3.65 μm, width of 400 nm for the n-region and 395 nm for the p-region, angular grating scatter size of 80 nm, and scatter order of 33. The scatter size for the absorption layer gradually increases from 80 nm to 180 nm. The excitation is identical to that in the homogeneous nanoribbon. The gap between rings in all samples is 100 nm in design.

The samples were fabricated using standard nanofabrication techniques. Hydrogen silsesquioxane (HSQ) solution in methyl isobutyl ketone (MIBK) (2:3 ratio) was used as negative electron beam lithography resist, spin-coated onto SOI wafer. The ratio of HSQ (FOX15) and MIBK was adjusted such that after exposure and development the resist was sufficiently thick as an etching mask for the subsequent etching process. The resist was then soft-baked and the structure was written by electron beam exposure. Electrons convert HSQ resist to an amorphous oxide. The patterned wafer was then developed in tetramethylammonium hydroxide (TMAH) solution (MFCD-26) at 60 °C for 40 sec and rinsed in DI water for 30 sec. The exposed and developed HSQ served as a mask for the subsequent reactive ion etching process that uses $SF_6/C_4F_8$ plasma. After dry etching, 3-μm thick silicon nitride was deposited by plasma-enhanced chemical vapor deposition (PECVD).

### Optical characterization

The characterization setup is illustrated in Supplementary Fig. 18. For cross-polarization reflection measurement, we excite the sample with linearly polarized monochromatic light at a broadband of angles and detect reflected light with an analyzer aligned orthogonal to the incident polarization. As a result of cross-polarization filtering, we can detect non-zero reflection only when the lattice Bloch mode has polarization oblique (i.e., neither identical nor orthogonal) to the incident polarization. The first aperture (the one closest to the tunable laser) is used during cross-polarization reflection measurements to confine the incident angle to within the first Brillouin zone, so that there is no overlap between signals from different Brillouin zones. For the characterization of specific band touching points (Fig. 2f–i), this aperture is used to allow only incident angles near the band touching points to pass through to the sample. The half-wave plate (HWP) was used as a convenient way to perform simultaneous rotation of the incident polarization and the effective analyzer polarization.

For characterization of the transport, incident polarization is aligned with the grating coupler (horizontal), and the analyzer is vertical, which is most efficient for the detection of the $(1,1)^T$ pseudospin and to suppress noise from the direct scattering of the incident light. Even with this cross-polarization filtering, we still observe scattering of stray light (possibly from light leaking out of the MMIs and subsequently scattered by the rings) that is especially strong in the first Brillouin zone. To mitigate this noise introduced by the stray light, we perform Fourier space imaging in the hexagonal Brillouin zone that is directly above the first Brillouin zone. The second aperture (the one in the image plane) is used during the transport characterizations to allow only a specific region from the nanoribbon to be imaged in the Fourier plane. During the characterization of the p-n junction (Fig. 4 and Supplementary Fig. 15), the aperture (rectangular, consisting of an adjustable slit and two knife edges) was used to allow only n- and p-regions to be imaged.

## Data availability

All data are available in the main text or the supplementary information files. All raw data generated during the current study are available from the corresponding authors upon request.

## Code availability

The FEM simulation models can be provided by the corresponding authors upon request.

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

## Acknowledgements

The authors thank B. Han for helpful discussions. We would like to acknowledge the support from National Science Foundation (NSF) (ECCS-1846766, CMMI-2037097). This work was partially supported by the NSF through the University of Pennsylvania Materials Research Science and Engineering Center (MRSEC) (DMR-1720530) and the University of Pennsylvania Center for Precision Engineering for Health (CPE4H). 3D band structure visualization printed courtesy of the University of Pennsylvania Libraries' Holman Biotech Commons.

## Author contributions

Z.G., C.-K.C. and L.F. conceived the project. C.-K.C., Z.G. and T.W. developed the theoretical model. Z.G., H.Z., Z.Z. conducted numerical simulations. H.Z., X.F. and X.Q. fabricated the samples. Z.G. and Z.Z. performed the characterization. L.F. guided the research. All authors contributed to discussions and manuscript preparation.

## Competing interests

The authors declare no competing interests.
