## [Peer Review File · Nature Communications]

Reviewers' Comments:

Reviewer #1:

Remarks to the Author:

The manuscript entitled "Discovery of a graphene sibling: the topological quadratic-node semimetal" reports an experimental demonstration of a symmetry-protected quadratic nodal point using a photonic ring lattice. The quadratic nodal point at the Brillouin zone center is doubly charged and pairs with two Dirac points at valleys. The manuscript and its supplementary material contain plentiful experimental data that support the main claim. However, given the high standard of Nature Communications, I do not think that this manuscript has novelty and impact strong enough to justify the acceptance.

My major concern is the following; the higher order Dirac (or Weyl in 3D cases) points enforced by spatial symmetries have been studied before in topological photonics, acoustics, etc. Some of them share similar features such as the quadratic dispersion or double topological charge. What is the difference between the quadratic node in this study and the previously reported higher-order Dirac points? What is the fundamental difference?

The authors emphasize the cross-spin coupling as the key ingredient. I recommend the authors to elaborate on it in more detail, especially since this cross-spin coupling differentiates this work from the previous publications. Why does it appear and what underpins it?

Could you explain the implication of the transportation of massive and massless chiral particles?

Ex. How this feature can be useful in photonic applications?

More detailed explanation of the spin momentum locking is needed.

While graphene is a kind of a signature that possesses nodal points in its bandstructure, I am not sure if it is appropriate to refer to this quadratic-node semimetal as a "graphene sibling".

Reviewer #2:

Remarks to the Author:

The paper by Gao and co-workers shows the experimental realization of an optical analogue of a quadratic semimetal, where linear Dirac cones and quadratic band touching points are located respectively in the $K-K'$ and in the Γ point of the Brillouin zone.

This is achieved in a triangular lattice of ring resonators, where the direction of rotation inside the ring resonator is associated to a pseudo-spin that is coupled within neighbouring sites. By expanding the Hamiltonian near the $K-K'-\Gamma$, winding numbers (topological charges) associated to these nodal points are theoretically found and experimentally observed via a polarization-resolved measurement in momentum space.

The main result of the paper is the demonstration of unique transport properties related to these nodal points, namely a hybrid superposition of massive and massless particles yielding simultaneous Klein and anti-Klein tunneling.

I think that the paper will have an impact for researchers working in nodal semimetal analogue, at the cross section of condensed matter physics and photonic systems. The observation of the topological charge of nodal points is also relevant for the topological photonics community, and may provide insights on the relation between polarization singularities (vortexes) and the Berry phase.

Moreover, the result about the superposition of massive and massless chiral particles is quite nice and interesting, because while switching from Klein to anti-Klein tunneling is possible in bilayer graphene or spin-orbit coupled systems, the coexistence of the two is unprecedented and may allow for more tunability in optical "valleytronic" effects for two/three different valley channels.

Overall, the paper is clearly written and well organized, the results are supporting the conclusions, which are sound and convincing. The supplemental materials is detailed and comprehensive, allowing for a good understanding of the work.

For these reasons, I think that the paper deserves publication in Nature Communications and I recommend publication as it is.

Reviewer #3:

Remarks to the Author:

This manuscript by Z. Gao et al. reports an experimental work on the observation of a novel kind of topologically protected band touching points. Contrary to Dirac cones that present linear dispersions and unitary winding numbers, these novel points located at the center of the Brillouin zone present a quadratic dispersion and ± 2 winding numbers. Moreover the authors have shown unconventional transport properties specific to these eigenstates, namely anti-Klein tunneling. To do this, they have realized a triangular array of ring resonators where clockwise and counter-clockwise propagating modes represent ± 1 pseudo-spin whose coupling depends both on the angular momentum index (l) and the in-plane momentum. I believe this is a very nice work, well presented and appropriately supported by experimental data. I would most likely recommend its publication, provided that the authors address the following points:

1- The main point I would like the authors to address regards the presentation of their experimental data. I feel that several points would deserve a more thorough data presentation to unambiguously support the conclusions. Firstly, in Fig. 2f-i, the authors present only two orthogonal polarizations. Although this strongly indicates a winding of 2, I would have appreciated to see a full polarization tomography in order to fully appreciate the winding of the wavefunction. At least numerically. This is particularly true for the Dirac cones where the two lobes are definitely not as clearly visible. Secondly, in Fig. 3 e-f, it is not clearly explained how the position of the green stars is defined. One definitely sees emission at other points in the BZ, but this is not reported in this figure. I would appreciate a clear protocol on how these points are extracted.

2- There are some minor aspects in the Fourier space images that I would suggest to clarify. Firstly, in Fig. 2 f-g, there is emission emerging from other BZ that does not show the appropriate polarization pattern; why is it so? In Fig. 3c, there is an asymmetry in the emission pattern along the y direction; how can this be explained? Finally, in Fig. 4 d and g, the emission at Gamma shows two spots, what is the physical origin of these two spots?

3- At line 98-102, it is mentioned that the Hamiltonian is PT-symmetric. I find this claim a bit misleading, as PT-symmetry is usually understood in the context of non-hermitian systems with gain and losses, where a device is not symmetric under P or T but is symmetric under the simultaneous action of P and T. Here, this is not the case, and this type of Hamiltonian is usually referred solely as chiral symmetric (as the authors do after). Indeed, under their assumptions, one could claim that normal graphene is PT symmetric which is definitely not how this terminology is usually defined. To avoid confusion, I would refrain from referring to PT-symmetry altogether.

4- The authors use a mapping (hence a coupling) between light direction of propagation (i.e. CW or CCW) and polarization. It would be highly relevant to explain how (or if) polarization affects the coupling and/or confinement energies, as this could play an important role in the behavior of light.

5- At line 196, the authors mention that they can inject a linear superposition of massless and massive particles (i.e. linearly and quadratically dispersive particles) simultaneously. Although this is indeed an important outcome of their work, they should explain clearly that this is only approximately true, as the Dirac cones and quadratic points are not exactly degenerate as is clearly visible in Fig. 3 e-f.

Response to Review Comments

Reviewer 1

The manuscript entitled “Discovery of a graphene sibling: the topological quadratic-node semimetal” reports an experimental demonstration of a symmetry-protected quadratic nodal point using a photonic ring lattice. The quadratic nodal point at the Brillouin zone center is doubly charged and pairs with two Dirac points at valleys. The manuscript and its supplementary material contain plentiful experimental data that support the main claim. However, given the high standard of Nature Communications, I do not think that this manuscript has novelty and impact strong enough to justify the acceptance.

We thank the reviewer for the positive view of the experimental data and the main claim, and we will address the question regarding novelty and impact in the following.

My major concern is the following; the higher order Dirac (or Weyl in 3D cases) points enforced by spatial symmetries have been studied before in topological photonics, acoustics, etc. Some of them share similar features such as the quadratic dispersion or double topological charge. What is the difference between the quadratic node in this study and the previously reported higher-order Dirac points? What is the fundamental difference?

We appreciate the reviewer's insightful comments on the higher-order Dirac and Weyl points that have been previously studied in photonic and acoustic topological metamaterials. Some of them indeed share similar local features compared to our quadratic-node semimetal, including quadratic dispersion and topological charge of two. However, our work presents a distinct nodal system when considering all nodes within the entire Brillouin Zone (BZ), setting it apart from any previously reported condensed-matter materials, as well as photonic or acoustic metamaterials. We have summarized their differences in the table below, which can be found in Supplementary Information Section 12 as Table S1. The quadratic-node semimetal reported here is the first 2D system featuring coexisting charge-2 and charge-1 nodes. The coexistence of quadratic and Dirac nodes, rooted in the cross-spin coupling and the commutation relations between symmetry operators, manifests profoundly the unique transport that we report here. For example, the coexisting massive and massless components in a chiral quasiparticle (depicted in Fig. 3), the simultaneous Klein and anti-Klein tunneling (shown in Fig. 4), and the unique inter-valley dynamics (presented in Supplementary Information Section 2 and Videos S4 and S5) are all tied to this unique nodal configuration. We believe this fundamental difference in nodal configuration, with profound impacts on transport properties, can sufficiently distinguish the quadratic-node semimetal reported here from any previously studied materials and metamaterials. In the revised manuscript, we have underscored this distinction and included references to prior research on quadratic nodes in photonic and acoustic systems to offer a more comprehensive context.

In addition to this critical difference in nodal configuration, the quadratic-node semimetal represents the first 2D example with a quadratic node obeying the Nielsen-Ninomiya theorem. Specifically, our clockwise (CW) and counterclockwise (CCW) modes in the experimental setup exhibit only two bands. Due to space-time inversion symmetry, only in a **two-band** model the fragile topology [1] can arise and quantize charges of nodes across the entire BZ. Consequently,

in our system, the total node charges are neutralized, as dictated by the Nielsen-Ninomiya theorem. This fundamental feature, associated with a quadratic node, is reported for the first time in our manuscript. Let us draw comparisons with existing literature. Firstly, we acknowledge that photonic crystal systems with charge-2 quadratic degeneracies are theoretically reported in [2] and preserve space-time inversion symmetry. Despite the local node being described by an effective two-band model with protection from fragile topology, this two-band model cannot be extended throughout the entire BZ, rendering the Nielsen-Ninomiya theorem inapplicable. Secondly, the experimentally observed charge-2 band touching in photonic crystals, as exemplified in [3,4], is not topologically protected. This is due to the presence of a nearby third band, which can approach the degeneracy and open a gap, demonstrating the fragility of the charge-2 node's topology [1]. Additionally, the quadratic degeneracy in [3,4] intersects with other bands, meaning these systems are not semimetals. To sum up, our system is described by an elegant two-band tight-binding model, much like the model of graphene except for the different nodal configuration, which is governed by the Nielsen-Ninomiya theorem. The theorem implies that the quadratic node is accompanied by two Dirac nodes. In contrast, previously reported photonic crystal systems are described by the limited low-energy theory valid only locally near the degeneracy points, making their physics distinct from ours. These distinctions, including system symmetries, the applicability of a two-band model, the validity of the Nielsen-Ninomiya (no-go) theorem, the existence and stability of the charge-2 and charge-1 nodes, are all summarized in Table S1.

In summary, we have revised the manuscript to place greater emphasis on the differences in nodal systems and the coexistence of charge-2 and charge-1 nodes in our quadratic-node semimetal, rather than focusing solely on the charge-2 quadratic node. We have also added Table S1 and Supplementary Information Section 12 to clearly distinguish our system from previously reported systems. We believe these modifications can sufficiently show the novelty of our system and address the reviewer's concern.

System	C_2T	Chiral	2-band	No-go theorem	Charge-2 node	Charge-1 node
(2D) Quadratic-node semimetal	Yes	Yes for weak coupling	Yes	Yes	Stable, fragile	Stable
(2D) Graphene	Yes	Yes	Yes	Yes	Absence	Stable
(2D) Gated bilayer graphene [5]	No	No	No	No	Unstable	Absence
(2D) PhC pillar array [2]	Yes	No	Locally	No	Stable, fragile	Absence
(2D) PhC hole array [3,4]	Yes	No	Locally	No	Stable, fragile	Absence
(3D) Weyl semimetal [6]	No	No	No	Yes	Absence	Stable
(3D) Weyl semimetal [7]	No	No	No	Yes	Stable	Stable

Table S1. Comparison between the quadratic-node semimetal and other example systems with linear or high-order nodes. The columns summarize whether the system possess space-time inversion (C_2T) symmetry, chiral symmetry, can be described by a 2-band model, obeys the Nielsen-Ninomiya (no-go) theorem, and the existence and stability of the charge-2 and charge-1 nodes.

The authors emphasize the cross-spin coupling as the key ingredient. I recommend the authors to elaborate on it in more detail, especially since this cross-spin coupling differentiates this work from the previous publications. Why does it appear and what underpins it?

We thank the reviewer for this insightful comment. The cross-spin coupling indeed differentiates this work from the previous publications. In short, the cross-spin coupling is a result of momentum (or equivalently, wavevector β) or power flow conservation during the coupling process between adjacent rings. Intuitively, this is depicted in Fig. 1b by the green and red arrows indicating power flow directions. In the following, we estimate the ratio between cross-spin coupling and same-spin coupling analytically to gain insight into the physics that underpins it and show full-wave simulation that agrees with the analytical approximation.

We calculate the coupling coefficient by evaluating the mode overlap between adjacent rings [8]. By abstracting the coupling region as two parallel waveguides (see Fig. S19a in the revised Supplementary Information), we can estimate the mode overlap (for example, the overlap between H_z components in a 2D model). Because our goal is to derive the ratio between cross-spin coupling and same-spin coupling, we drop some of the normalization factors and write the mode overlap as

$$\begin{aligned}\kappa &\sim \frac{1}{2\pi R} \int_0^L (e^{i\beta_1 l} e^{-\gamma d})^* e^{i\beta_2 l} dl \\ &= -\frac{ie^{-\gamma d}}{2\pi R} \frac{e^{i(\beta_2 - \beta_1)L} - 1}{\beta_2 - \beta_1}\end{aligned}$$

where $\beta_{1,2}$ are the wavevectors in the left and right waveguides, respectively, and they can be of the same sign (for cross-spin coupling) or opposite signs (for same-spin coupling). L is the length of the coupling region, d is the gap between waveguides, γ is the decay rate of the field amplitude in the cladding, and l is the local coordinate along the waveguide. For cross-spin coupling, we have $\beta_1 = \beta_2$ and $|\kappa| = \frac{e^{-\gamma d L}}{2\pi R}$. For same-spin coupling, we have $\beta_1 = -\beta_2 = \beta$ and $|\kappa| = e^{-\gamma d} \frac{|e^{2i\beta L} - 1|}{4\pi R \beta}$. The amplitude ratio between same-spin and cross-spin coupling is therefore

$$\frac{|\kappa_{same-spin}|}{|\kappa_{cross-spin}|} = \frac{|e^{i2\beta L} - 1|}{2\beta L}$$

This model assumes adiabatic transition into and out of the coupling region and does not capture the non-adiabatic nature of transition when the radius of the microring is very small. However, it intuitively shows that the momentum conservation in cross-spin coupling manifest as the linear increase of coupling strength with longer coupling length L . On the other hand, the same-spin coupling has momentum mismatch of 2β and the coupling strength oscillates around a constant. Therefore, when the WGM mode order is large, the long coupling region ensures that the cross-spin coupling dominates. We perform full-wave simulation using COMSOL and show the ratio between same-spin and cross-spin coupling in Fig. S19. The decrease of same-spin coupling relative to cross-spin coupling is approximately $radius^{-2}$ in terms of power ratio. Since the effective coupling length scales approximately linearly with the radius of the microrings (Fig. S19c), this simulation result agrees with the analytical estimation.

In summary, we have added analytical and simulation results to show that the cross-spin coupling dominates over same-spin coupling as a result of momentum conservation during coupling. The ratio between same-spin and cross-spin coupling decreases approximately quadratically with the radius of the microring, and in our system, with a radius of 3.5 μm , this ratio is sufficiently small (about 0.3%). We have added the sentence "The cross-spin coupling is a result of momentum conservation during the coupling process between adjacent rings (depicted in Fig. 1b and quantitatively analyzed in Supplementary Information Section 11)." and Supplementary Information Section 11 in the revised manuscript and Supplementary Information, respectively.

Could you explain the implication of the transportation of massive and massless chiral particles?
Ex. How this feature can be useful in photonic applications?

The massive and massless chiral particles both exhibit spin-momentum locking, which is one of the characteristic traits of chiral particles. The spin-momentum locking in our systems is in direct analogy to the spin-momentum locking well known in topological insulators with the Rashba term [9]. The particle's spin determines the direction of its momentum. For massless chiral particles near a charge-1 Dirac node, because the spin of the eigenstate winds once around the Dirac node, the direction of momentum and the spin has 1-to-1 mapping. In other words, the spin uniquely determines the momentum direction. For massive chiral particles near a charge-2 quadratic node, because the spin of the eigenstate winds twice around the node, there are two possible momentum directions for each spin. This feature can enable numerous photonic applications in the future. For example, with spin-polarized injection on the edge of a photonic lattice, shown in Fig. 3, we can control the power flow direction by controlling the injection spin, which is a new mechanism for beam steering. The spin-momentum locking also dictates the transport at a p-n junction. At normal incidence, this is demonstrated in Fig. 4 as simultaneous Klein and anti-Klein tunneling. At oblique angles, we expect richer dynamics following the spin-momentum locking and the spin conservation. In photonics, with the capability of modulating the refractive index and hence the resonance frequencies of the resonators, we can expect distinct modulation response for the massless and massive chiral particles, which have the potential for communication, information processing, power transfer, and etc. With an extra node in our system compared to graphene, we can also expect the study of valleytronics in our system with an extra dimension, potentially useful for optical communications with the information encoded in the valley degree of freedom.

We have revised the paragraph under "Quantum transport in the quadratic-node semimetal" and the conclusion paragraph to elaborate on the implication of the transport of massive and massless chiral particles and potential photonic applications.

More detailed explanation of the spin momentum locking is needed.

We have added more explanation of the spin-momentum locking in the revised manuscript. Please see the paragraph under "Quantum transport in the quadratic-node semimetal".

While graphene is a kind of a signature that possesses nodal points in its bandstructure, I am not sure if it is appropriate to refer to this quadratic-node semimetal as a "graphene sibling".

Graphene is a signature material that possesses linear nodal points. By “graphene sibling”, we emphasize that the quadratic-node semimetal inherits graphene physics, with massless Dirac particle and Klein tunneling supported by its linear nodal points (the signature of graphene), but at the same time features a new dimensionality represented by its unique quadratic node. We think the analogy is appropriate in this sense. In addition, in Table S1, added in the revised manuscript, we can see that the quadratic-node semimetal has the same characteristics as the graphene except for the charge-2 node. On the other hand, both the quadratic-node semimetal and the graphene have very distinct characteristics compared to other systems in the table. It is clear from the table that graphene and the quadratic-node semimetal resemble each other while being distinct from other systems. We appreciate this comment by the reviewer, and to address this concern, we have added these explanations to the revised manuscript, in the last paragraph in the introduction section and in the conclusion paragraph.

Reviewer #2:

The paper by Gao and co-workers shows the experimental realization of an optical analogue of a quadratic-node semimetal, where linear Dirac cones and quadratic band touching points are located respectively in the K - K' and in the Γ point of the Brillouin zone.

This is achieved in a triangular lattice of ring resonators, where the direction of rotation inside the ring resonator is associated to a pseudo-spin that is coupled within neighbouring sites. By expanding the Hamiltonian near the K - K' - Γ , winding numbers (topological charges) associated to these nodal points are theoretically found and experimentally observed via a polarization-resolved measurement in momentum space.

The main result of the paper is the demonstration of unique transport properties related to these nodal points, namely a hybrid superposition of massive and massless particles yielding simultaneous Klein and anti-Klein tunneling.

I think that the paper will have an impact for researchers working in nodal semimetal analogue, at the cross section of condensed matter physics and photonic systems. The observation of the topological charge of nodal points is also relevant for the topological photonics community, and may provide insights on the relation between polarization singularities (vortexes) and the Berry phase.

Moreover, the result about the superposition of massive and massless chiral particles is quite nice and interesting, because while switching from Klein to anti-Klein tunneling is possible in bilayer graphene or spin-orbit coupled systems, the coexistence of the two is unprecedented and may allow for more tunability in optical “valleytronic” effects for two/three different valley channels.

Overall, the paper is clearly written and well organized, the results are supporting the conclusions, which are sound and convincing. The supplemental materials is detailed and comprehensive, allowing for a good understanding of the work.

For these reasons, I think that the paper deserves publication in Nature Communications and I recommend publication as it is.

We thank the Reviewer for the positive comments.

Reviewer #3:

This manuscript by Z. Gao et al. reports an experimental work on the observation of a novel kind of topologically protected band touching points. Contrary to Dirac cones that present linear dispersions and unitary winding numbers, these novel points located at the center of the Brillouin zone present a quadratic dispersion and ± 2 winding numbers. Moreover the authors have shown unconventional transport properties specific to these eigenstates, namely anti-Klein tunneling. To do this, they have realized a triangular array of ring resonators where clockwise and counter-clockwise propagating modes represent ± 1 pseudo-spin whose coupling depends both on the angular momentum index (l) and the in-plane momentum. I believe this is a very nice work, well presented and appropriately supported by experimental data. I would most likely recommend its publication, provided that the authors address the following points:

We thank the reviewer for the positive view on our work, and we will address the questions in the following.

1- The main point I would like the authors to address regards the presentation of their experimental data. I feel that several points would deserve a more thorough data presentation to unambiguously support the conclusions. Firstly, in Fig. 2f-i, the authors present only two orthogonal polarizations. Although this strongly indicates a winding of 2, I would have appreciated to see a full polarization tomography in order to fully appreciate the winding of the wavefunction. At least numerically. This is particularly true for the Dirac cones where the two lobes are definitely not as clearly visible. Secondly, in Fig. 3 e-f, it is not clearly explained how the position of the green stars is defined. One definitely sees emission at other points in the BZ, but this is not reported in this figure. I would appreciate a clear protocol on how these points are extracted.

In addition to the two polarizations shown in Figs. 2f-i, we have characterized the Fourier-space images with continuous (every 10 degrees for the quadratic cone and every 5 degrees for the Dirac cone) rotation of polarization orientation, shown in Supplementary Videos S1 and S2. In these two videos, the dark lines show continuous rotation and provide unambiguous visual confirmations of the winding of the wavefunction. The rotation angle of the dark lines relative to the polarization rotation ($1x$ for the quadratic cone and $2x$ for the Dirac cone) again confirms the winding numbers. Numerically, the continuous polarization winding for the wavefunction is shown in Figs. 2d&e. We believe the comparison between the numerical results in Figs. 2d&e and the experimental results in Supplementary Videos S1 and S2 show convincing evidence regarding the winding of the wavefunction beyond the two orthogonal polarizations in Fig. 2f-i. The limited signal-to-noise ratio, especially in Fig. 2h&i, as the reviewer has pointed out that, is due to the limited excitation efficiency and the imperfect filtering of the directly reflected incident beam. When we use free-space plane waves to excite the resonant modes in the ring array, the excitation efficiency is low as a result of poor modal overlapping, since the ring array has large void areas inside and between the rings. The majority of the incident beam does not excite the modes inside the photonic lattice and the direct reflection and diffraction of this incident beam becomes stray light. We filter out most of this stray light with cross-polarization filtering. However, this polarization filtering is perfect only at normal incidence. Hence the polarization filtering works best at the center of the

first BZ, and consequently the Dirac cones at the corner of the BZ have lower signal-to-noise ratio due to imperfect filtering of the stray light. We have added these clarifications to the caption of Fig. 2 and the Supplementary Information Section 8.

For Figs. 3e-f, the positions of the green stars are defined as the positions of the intensity peak(s) in the Fourier-space images along the $k_y = 0$ line cut (see updated Fig. S17). The protocol is to extract the highest intensity peak first. If there is a second peak visible from the Fourier-space image, then we also plot the second peak. Because of the limited signal-to-noise ratio (here the noise also includes stray light scattered from the input grating coupler and the MMIs), it is easier to distinguish the real signal from the stray light in the 2D Fourier-space images than in the 1D line cuts. Identification of the “second highest peak” may not be very obvious due to limited signal-to-noise ratio. However, for the two wavelengths shown in Fig. 3d, the second highest peaks can be clearly identified, especially when comparing with the simulated Fourier-space images. Hence, we choose to plot these second-highest peaks as well. To make this protocol clearer, we have changed the symbols for the second-highest peaks to hollow green stars, while the symbols for the highest peaks are solid-filled green stars. We note that the quantitative agreement between the experimental and simulated Fourier-space images are obvious, as shown in Figs. 3c&d and Figs. S17a&b. The green stars in Figs. 3e&f are plotted to help connecting the Fourier-space images measured at discrete excitation wavelengths to the dispersion relationship. The agreement between the experimental result and the simulation is unambiguously demonstrated by the 2D Fourier-space images in Figs. 3c&d and Fig. S17a&b. The extracted solid (hollow) green stars are rigorously defined as the highest (second highest) peaks and showing them along the simulated dispersion curve serves as supportive evidence. We have updated Fig. 3, Fig. S17, their captions, and Supplementary Information Section 9 to show this clearer protocol.

2- There are some minor aspects in the Fourier space images that I would suggest to clarify. Firstly, in Fig. 2 f-g, there is emission emerging from other BZ that does not show the appropriate polarization pattern; why is it so? In Fig. 3c, there is an asymmetry in the emission pattern along the y direction; how can this be explained? Finally, in Fig. 4 d and g, the emission at Gamma shows two spots, what is the physical origin of these two spots?

For the first question, the emission emerging from other BZ contains more stray light (direct diffraction of the incident beam) because the cross-polarization filtering that is responsible for eliminating the stray light performs best only at normal angle. The appropriate polarization patterns are buried underneath the stray light, making them difficult to observe. Diffraction can be understood as the microring lattice providing in-plane momentum $\mathbf{G} = m_1\mathbf{b}_1 + m_2\mathbf{b}_2$ to diffract the incident beam to other BZs in the far field image. Here $\mathbf{b}_{1,2}$ are the primitive wavevectors of the reciprocal lattice. At BZs other than the center one, the diffraction angle slightly rotates the polarization and adds nonzero component to the orthogonal polarization direction, which will not be filtered out by the analyzer and consequently appears in the Fourier-space images. However, this polarization rotation does not happen when the polarization orientation is orthogonal to or aligned with the in-plane momentum \mathbf{G} . Hence, we do observe the appropriate polarization patterns at other BZs when the polarizer and analyzer orientations happen to be aligned with the corresponding \mathbf{G} . This can be observed, for example, in Supplementary Video S1 when polarizer is at 120° and 240° . We appreciate this thoughtful question from the reviewer and have added this explanation to the caption of Fig. 2 and the Supplementary Information Section 8.

For the second question regarding Fig. 3c, the asymmetry along the y direction can be explained by the Fourier-space emission profile of the individual microring resonators. Most of the discussions in the manuscript focus on the fine structures in the Fourier-space images, which reveals the wavefunctions of the Bloch waves in the lattice. However, the large-scale envelope the Fourier space image is determined by the emission profile of the individual microrings, shown in the Supplementary Information Section 6 and Fig. S13. The emission profile of mode order 35 is stronger in the lower half of this particular BZ (which is directly above the first BZ), and this is consistent with what we observe in Fig. 3c. Again, we thank the reviewer for this excellent question, and we have added our explanation to the caption of Fig. 3 and the Supplementary Information Section 6.

For the last question regarding Fig. 4 d and g, the two spots near Gamma point correspond to the incident and the reflected waves (with group velocities in $+x$ and $-x$ directions), respectively. Because the band-touching near the Gamma point is quadratic, we observe anti-Klein tunneling, which features strong reflection at the p-n junction. The fact that we see two spots agrees very well with the theory and directly shows that the quantum transport in the Gamma valley can be modulated very differently compared to the Dirac valleys. We have added this further explanation to the caption of Fig. 4.

3- At line 98-102, it is mentioned that the Hamiltonian is PT-symmetric. I find this claim a bit misleading, as PT-symmetry is usually understood in the context of non-hermitian systems with gain and losses, where a device is not symmetric under P or T but is symmetric under the simultaneous action of P and T. Here, this is not the case, and this type of Hamiltonian is usually referred solely as chiral symmetric (as the authors do after). Indeed, under their assumptions, one could claim that normal graphene is PT symmetric which is definitely not how this terminology is usually defined. To avoid confusion, I would refrain from referring to PT-symmetry altogether.

We thank the reviewer for this thoughtful comment, and we understand the reviewer's concern. We have changed the term to space-time inversion symmetry to avoid this confusion.

4- The authors use a mapping (hence a coupling) between light direction of propagation (i.e. CW or CCW) and polarization. It would be highly relevant to explain how (or if) polarization affects the coupling and/or confinement energies, as this could play an important role in the behavior of light.

We agree with the reviewer about the importance of this mapping and the role of polarization in general. This mapping between power flow direction and the polarization of emission can play an important role when the photonic lattice forms coupling with other structures, for example, another metasurface, 2D photonic lattice, or 2D material, to form bi-layer structures. This mapping can be freely controlled by the angular locations of the scatters, which is an advantage of our system. When the number of scatters per resonator (i.e., grating order) does not equal to the mode order, this change of mapping would not affect the light flow (or effective Hamiltonian) within the lattice. However, if the grating order matches the mode order, then there exist off-diagonal terms in the on-site energy, which represents the cross-spin coupling (i.e., spin-flipping process) within a single ring. We have added this description to the manuscript and Supplementary Information Section 6.

More detailed study is certainly interesting, and we would like to address this interesting behavior and the flexible control related to this in future publications.

5- At line 196, the authors mention that they can inject a linear superposition of massless and massive particles (i.e. linearly and quadratically dispersive particles) simultaneously. Although this is indeed an important outcome of their work, they should explain clearly that this is only approximately true, as the Dirac cones and quadratic points are not exactly degenerate as is clearly visible in Fig. 3 e-f.

The reviewer is correct. The Dirac and quadratic cones are not exactly degenerate, and there are certain frequency ranges where we cannot inject both massless and massive components. We have added a clear note to this in the caption of Fig. 3: “This (the energy offset) reduces the energy range within which we can inject both massive and massless components, as compared to the tight-binding model calculations in Fig. 1 and Figs. S3-10”. However, we want to emphasize that although the cones are not exactly degenerate, we can still find an energy range where we can inject a linear superpositions of massless and massive particles, as shown in Fig. 4. More importantly, this offset between linear and quadratic band touching energies is due to the strong coupling strength we choose to overcome the frequency detuning due to fabrication imperfections limited strongly by the fabrication facility. This is not a fundamental limit to the system itself, as we show this energy offset is negligible when the coupling is weak (Fig. S11a&b).

- [1] T. Bzdušek and M. Sigrist, *Robust Doubly Charged Nodal Lines and Nodal Surfaces in Centrosymmetric Systems*, Phys. Rev. B **96**, 155105 (2017).
- [2] Y. D. Chong, X.-G. Wen, and M. Soljačić, *Effective Theory of Quadratic Degeneracies*, Phys. Rev. B **77**, 235125 (2008).
- [3] Y. Zhang, A. Chen, W. Liu, C. W. Hsu, B. Wang, F. Guan, X. Liu, L. Shi, L. Lu, and J. Zi, *Observation of Polarization Vortices in Momentum Space*, Phys. Rev. Lett. **120**, 186103 (2018).
- [4] A. Chen, W. Liu, Y. Zhang, B. Wang, X. Liu, L. Shi, L. Lu, and J. Zi, *Observing Vortex Polarization Singularities at Optical Band Degeneracies*, Phys. Rev. B **99**, 180101 (2019).
- [5] J. B. Oostinga, H. B. Heersche, X. Liu, A. F. Morpurgo, and L. M. K. Vandersypen, *Gate-Induced Insulating State in Bilayer Graphene Devices*, Nat. Mater. **7**, 151 (2008).
- [6] S.-Y. Xu et al., *TOPOLOGICAL MATTER. Discovery of a Weyl Fermion Semimetal and Topological Fermi Arcs*, Science **349**, 613 (2015).
- [7] S.-M. Huang et al., *New Type of Weyl Semimetal with Quadratic Double Weyl Fermions*, Proc. Natl. Acad. Sci. **113**, 1180 (2016).
- [8] J.-S. Hong, *Couplings of Asynchronously Tuned Coupled Microwave Resonators*, IEE Proc. - Microw. Antennas Propag. **147**, 354 (2000).
- [9] M. Z. Hasan and C. L. Kane, *Colloquium: Topological Insulators*, Rev. Mod. Phys. **82**, 3045 (2010).

Reviewers' Comments:

Reviewer #1:

Remarks to the Author:

The authors have well addressed my comments. The revised manuscript is clear. I now recommend this manuscript be published in Nature Communications.

Reviewer #3:

Remarks to the Author:

The authors have well addressed my main comments. I now recommend the publication of their manuscript.